# *Adlercreutzia equolifaciens* Is an Anti-Inflammatory Commensal Bacterium with Decreased Abundance in Gut Microbiota of Patients with Metabolic Liver Disease

**DOI:** 10.3390/ijms241512232

**Published:** 2023-07-31

**Authors:** Florian Plaza Oñate, Célia Chamignon, Sebastian D. Burz, Nicolas Lapaque, Magali Monnoye, Catherine Philippe, Maxime Bredel, Laurent Chêne, William Farin, Jean-Michel Paillarse, Jérome Boursier, Vlad Ratziu, Pierre-Yves Mousset, Joël Doré, Philippe Gérard, Hervé M. Blottière

**Affiliations:** 1Université Paris-Saclay, INRAE, MGP, MetaGenoPolis, 78350 Jouy-en-Josas, France; florian.plaza-onate@inrae.fr (F.P.O.); joel.dore@inrae.fr (J.D.); 2NovoBiome, 33360 Latresne, France; c.chamignon@novobiome.eu (C.C.); m.bredel@novobiome.eu (M.B.); py.mousset@novobiome.eu (P.-Y.M.); 3Université Paris-Saclay, INRAE, AgroParisTech, Micalis Institute, 78350 Jouy-en-Josas, France; sebastian.burz@unil.ch (S.D.B.); nicolas.lapaque@inrae.fr (N.L.); magali.monnoye@inrae.fr (M.M.); philippe.gerard@inrae.fr (P.G.); 4Enterome, 75011 Paris, France; lchene@enterome.com (L.C.); wfarin@enterome.com (W.F.); jmpaillarse@enterome.com (J.-M.P.); 5Université d’Angers, SFR ICAT4208, Laboratoire HIFIH & Centre Hospitalier d’Angers, 49100 Angers, France; jeboursier@chu-angers.fr; 6Sorbonne-Université, Hôpital Pitié-Salpêtrière, INSERM UMRS 1138, Centre de Recherche des Cordeliers, 75006 Paris, France; vlad.ratziu@inserm.fr; 7Nantes-Université, INRAE, UMR 1280, PhAN, 44000 Nantes, France

**Keywords:** NAFLD, gut microbiota, inflammation, metagenomics, live biotherapeutic product

## Abstract

Non-alcoholic fatty liver disease (NAFLD) affects about 20–40% of the adult population in high-income countries and is now a leading indication for liver transplantation and can lead to hepatocellular carcinoma. The link between gut microbiota dysbiosis and NAFLD is now clearly established. Through analyses of the gut microbiota with shotgun metagenomics, we observe that compared to healthy controls, *Adlercreutzia equolifaciens* is depleted in patients with liver diseases such as NAFLD. Its abundance also decreases as the disease progresses and eventually disappears in the last stages indicating a strong association with disease severity. Moreover, we show that *A. equolifaciens* possesses anti-inflammatory properties, both in vitro and in vivo in a humanized mouse model of NAFLD. Therefore, our results demonstrate a link between NAFLD and the severity of liver disease and the presence of *A. equolifaciens* and its anti-inflammatory actions. Counterbalancing dysbiosis with this bacterium may be a promising live biotherapeutic strategy for liver diseases.

## 1. Introduction

Non-alcoholic fatty liver disease (NAFLD) defines a spectrum of liver injury ranging from the early phases of steatosis or non-alcoholic fatty liver (NAFL) to the progressive form of non-alcoholic steatohepatitis (NASH), where hepatic inflammation can drive fibrosis progression to cirrhosis and hepatocellular carcinoma. This condition is strongly linked to unhealthy diets and sedentary lifestyles, all very prevalent in industrial societies. NAFLD, which is the hepatic manifestation of the metabolic syndrome, is the leading cause of chronic liver diseases in Western countries as well as in other regions, with a prevalence as high as 30% in the general population [1] and an increasing incidence [2]. Although few preventive treatments are currently assessed, no approved pharmacological treatments to stop or limit the progression of NAFLD are currently available, so the management of the disease is mainly based on measures enforcing healthy diet and lifestyle [3].

Over the last decade, several reports have linked gut microbiota modifications to NAFL or NASH [4,5,6,7]. However, the results of studies on dysbiosis in NAFLD patients are divergent, and therefore, a clear gut microbiota signature in NAFLD patients is still a matter of debate. A meta-analysis of 13 published studies using 16S rRNA gene sequencing reported a decreased abundance of Bacteroidota and *Ruminococcaceae* associated with an increased abundance of *Veillonellaceae*, *Lactobacillaceae*, and *Dorea* [8]. However, another study, in an attempt to correlate the severity of the disease with gut microbiota shift, observed increased levels of *Bacteroides* that have been correlated with NASH, while an increased abundance of *Ruminococcus* was reported linked with fibrosis [9]. Similarly, using shotgun metagenomics, Loomba and colleagues characterized the gut microbiome composition of NAFLD patients and provided evidence for a gut microbiota signature specific to advanced fibrosis [10]. Moreover, our lab recently published data suggesting that nonvirulent endotoxin-producing strains of opportunistic pathogens, which are more abundant in the gut microbiota of obese patients (*Enterobacter cloacae*, *Escherichia coli*, and *Klebsiella pneumoniae*), may act as causative agents for the induction of NAFLD [11]. These results suggest that gut microbiota dysbiosis may exacerbate hepatic steatosis and control the rate of NAFLD progression. The underlying mechanisms are complex and not fully understood but several pathways have been highlighted. Notably, it has been shown that the microbiota promotes the storage of triglycerides in adipocytes through the suppression of the intestinal expression of a circulating lipoprotein lipase (LPL) inhibitor Angiopoietin-like 4 (ANGPTL4) [12,13]. Increased ethanol production [14,15] and the induction of a choline deficit by the microbiota could also promote triglyceride accumulation in the liver [16,17,18]. Among the potential mediators of this association, lipopolysaccharide (LPS) exerts relevant metabolic and proinflammatory effects [19]. Inflammation is considered a major contributor to NAFLD and is linked to increased NF-κB signaling pathway in both the intestine and liver [20].

Moreover, the farnesoid X receptor (FXR) pathway has been proposed as an important mediator of gut microbiota effect on lipid and glucose metabolism through secondary bile acids production [21]. Recent studies have also shown that the bacterial metabolites of branched-chain and aromatic amino acids, especially 3-(4-hydroxyphenyl) lactate and phenylacetic acid, were associated with steatosis and the development of fibrosis [22,23]. These and other studies from the literature clearly establish the major importance of a better understanding of the relationships between gut microbiota and NAFLD onset, which could also lead to new therapeutic approaches.

Fecal microbiota transfer (FMT) can be a way to demonstrate causality but also to treat patients suffering from liver diseases. As an example, FMT was reported to diminish hospital stays, improve cognition, and impact gut microbiota dysbiosis in cirrhotic patients with recurrent hepatic encephalopathy [24]. Furthermore, through manipulation of the gut microbiota in animal models, the direct role of the microbiome in the development of NAFLD has been highlighted [25,26]. Our laboratory revealed that the gut microbiota contributed to the development of steatosis induced by a high-fat diet in a murine model. Moreover, by using FMT from mouse to mouse, we also showed that the severity of the steatosis depends on the microbiota composition [27]. Transferring the microbiome of NAFLD-resistant mice to wild-type mice renders the latter resistant to NAFLD. Conversely, the transfer of the microbiome from mice susceptible to high-fat diet-induced NAFLD renders recipients sensitive to the diet. More recently, we confirmed that the gut microbiota contributes to steatosis development using human-to-mice FMT [28]. Thus, these data suggest that modulating the gut microbiota may be a new strategy to prevent or treat NAFLD and associated metabolic disorders.

In the present study, we show that the abundance of *Adlercreutzia equolifaciens*, an Actinobacteriota (recently renamed Actinomycetota) from the *Eggerthellaceae* family, is linked to a healthy status of the human microbiome and is reduced in patients with NAFLD. This depletion is also strongly linked to the severity of the liver disease, and its abundance is very low in cirrhosis, with almost disappearance in decompensated patients and in acute-on-chronic liver failure (ACLF). We also show that this bacterium possesses anti-inflammatory properties, both in vitro and in vivo, in a humanized mouse model of NAFLD. Based on these results, we propose that counterbalancing dysbiosis using *A. equolifaciens* as a live biotherapeutic product may be a promising strategy for the prevention and/or treatment of liver diseases.

## 2. Results

### 2.1. Decreased Abundance of A. equolifaciens in Microbiome of Liver Disease Patients NASH1 and NASH2

Shotgun metagenomics was performed on the fecal DNA of all patients in the NASH1 cohort—our discovery study. A non-significant reduction in global microbial diversity was observed with increasing histological severity. However, twenty-one MGS discriminating NASH (steatohepatitis with or without fibrosis/cirrhosis) from the normal/fatty liver (simple steatosis) population were identified with a combined approach of the Wilcoxon rank sum and Kolmogorov–Smirnov tests.

In a subsequent study on NASH2 conducted in 2012, 3 out of the 21 MGS previously identified in a NASH1 study were validated and yielded significant results (Wilcoxon rank sum test). We were able to establish a phylogenetic association between two of these MGS and known bacterial species. The first MGS was *Escherichia coli*, while the second one, which is the focus of the current study, was found to be *A. equolifaciens*, with 71.2% of the genes in MGS 9828_3 being annotated to *A. equolifaciens*. This particular species belongs to the Actinobacteriota phylum, now renamed Actinomycetota, and was initially isolated from fecal samples in Japan [29]. *A. equolifaciens* was detected in most of the healthy subjects, with its relative abundance decreasing as the severity of the disease increased (Figure 1).

Then, we downloaded publicly available shotgun metagenomic data from liver disease patients, and we performed taxonomic profiling to detect and estimate the abundance of *A. equolifaciens*. First, we considered a study from Loomba and colleagues comparing the gut microbiome of 85 patients with biopsy-proven NAFLD (1 sample with too few reads was filtered out), among which 71 had mild to moderate disease (fibrosis stages between 0 and 2), and 14 presented advanced fibrosis (stage 3 or 4) [10]. Remarkably, the prevalence and abundance of *A. equolifaciens* strongly decreased with the level of fibrosis (Figure 2). Unfortunately, the data published in this study did not include any healthy controls.

Mining a Chinese cohort of 112 liver cirrhosis patients and 102 healthy controls [30], we observed similar features, but with a more pronounced loss of *A. equolifaciens* in cirrhotic patients (Figure 3). The prevalence of this MGS was 57% in healthy controls, whereas it was only found in 16% of liver cirrhosis patients.

Finally, we looked at fecal microbiome data from 166 Spanish patients with liver cirrhosis [31] and compared them to 38 Spanish healthy controls of the MetaHIT project. Again, *A. equolifaciens* was strongly associated with the health status of donors and was significantly depleted in cirrhotic patients. The depletion of the bacterium was all the more pronounced as the disease progressed and reached higher stages of severity (Figure 4).

### 2.2. A. equolifaciens Encompasses Two Distinct Genomospecies with Strong Associations with Geography

Exploring several public databases, we retrieved 154 high-quality Metagenome-Assembled Genomes (MAGs) of *A. equolifaciens* and 11 genomes derived from isolates. All genomes came from fecal material, essentially from humans but also from mice and rats (154, 10, and 1, respectively). We computed Average Nucleotide Identity (ANI) between all pairs of genomes, and we performed average linkage hierarchical clustering (UPGMA). Finally, we extracted clusters with a 95% ANI cut-off widely used to delineate microbial species [32].

Interestingly, genomes were clustered into two phylogroups, defining two distinct genomospecies (Figure 5). The first phylogroup consisted of, among others, the *A. equolifaciens*-type strain (DSM 19450) as well as the type strain of the subspecies *celata* (DSM 18785). The second phylogroup was made up of the type strains of two recently validated species, *Adlercreutzia rubneri* [33] and *Adlercreutzia hattorii* [34]. Notably, the genomes of these two type strains were closely related (ANI = 97.2%), suggesting that *A. rubneri* and *A. hattorii* actually represent the same species.

Remarkably, we found a strong association between phylogroups and the geographical origin of the host (chi-squared test *p* < 2.2 × 10^−16^, Figure 6). Indeed, 82% of the genomes assigned to the first phylogroup were obtained from the fecal material of Asian individuals. In contrast, the second phylogroup mostly represents genomes recovered from the feces of European and North American individuals (92% of genomes).

Finally, we noticed that genes responsible for equol synthesis, a function that has given its name to this species, were frequently detected in genomes assigned to the first phylogroup but were rare in the second (60% vs. 8.8% of genomes, respectively, chi-squared test *p* < 3.7 × 10^−12^, Appendix A). In other words, this result highlights that equol-producing strains are much more prevalent in Asian individuals. Interestingly, in the second phylogroup, strains predicted as equol producers were found almost exclusively in mouse feces (9/10). Thus, the equol pathway could be vanishing in strains of the second phylogroup, but only in humans and not in mice.

### 2.3. A. equolifaciens Displays Anti-Inflammatory Properties In Vitro

We then used the strain DSM 19450 to look for its impact on the NF-κB inflammatory pathway using stably transfected reporter cell lines. As illustrated in Figure 7a, HT-29 NF-κB cells (colonic epithelial cells) used in the test responded strongly to TNF-α by an over 11-fold activation of the NF-κB pro-inflammatory pathway. A bacterial culture medium, M104, in the absence of TNF-α, was used as the negative control. When incubated with TNF-α in the presence of the supernatant of *A. equolifaciens*, the fold activation was reduced to 8 indicating a mild but significative inhibition of the NF-κB pathway. In contrast, the supernatant of *Parvibacter caecicola*, a closely related member of the *Eggerthellaceae* family, had no effect on NF-κB activation following TNF-α stimulation. The culture supernatant of both bacterial strains did not affect NF-κB activity in hepatocytic cells, HepG2, stimulated with TNF-α. Appendix A shows that the culture medium and the supernatant of both bacterial strains, *A. equolifaciens* and *P. caecicola*, had no effect on cell viability, as assessed by measuring MTS activity.

Figure 7b shows that HT-29 NF-κB cells used in the test responded strongly to TNF-α by an over 15-fold activation of the NF-κB pro-inflammatory pathway. PBS used to resuspend bacterial pellet in the absence of TNF-α was used as the negative control. When incubated with TNF-α in the presence of the pellet of *A. equolifaciens*, the fold activation was reduced to two, indicating a very strong inhibition of the NF-κB pathway. In contrast, the pellet of *P. caecicola* exerted a mild and non-statistically significant effect on NF-κB activation following TNF-α stimulation. Again, cell viability was not affected by both strains, as assessed by measuring MTS activity (Appendix A).

In Figure 7c, it is shown that HepG2 NF-κB cells responded strongly to TNF-α by a five-fold activation of the NF-κB pro-inflammatory pathway. PBS used to resuspend bacterial pellet in the absence of TNF-α was used as the negative control. When incubated with TNF-α in the presence of the pellet of *A. equolifaciens*, the fold activation was strongly reduced to a level close to non-activated cells. In contrast, the pellet of *P. caecicola* had no effect on NF-κB activation following TNF-α stimulation. Cell viability was not affected by the pellet of *A. equolifaciens* and *P. caecicola*.

### 2.4. A. equolifaciens Displays Anti-Inflammatory Effect In Vivo and Impacts Mice Metabolism

The effect of DSM 19450 *A. equolifaciens* strain given by oral gavage and used as a biotherapeutic agent was evaluated in a mouse model of hepatic steatosis, i.e., C57Bl/6J SPF male mice treated with antibiotics and then inoculated with the gut microbiota of patients with NASH and further exposed to a high-fat, high-fructose diet. Mice were force-fed daily with *A. equolifaciens* resuspended in PBS or only PBS as control. Food consumption during the 10 weeks of follow up was similar in the two groups. Interestingly, mice gavaged with *A. equolifaciens* gained less weight than the controls (Figure 8a,b). This persisted during the 10 weeks of the experiment. We also observed that mice that received *A. equolifaciens* displayed reduced hyperglycemia at 1 month, as compared to control mice (Figure 8c). However, when an oral glucose tolerance test (OGTT) was performed on day 70, after 1.5 months of induction and gavage, there was no longer any difference in fasting blood glucose or glucose tolerance between the two groups. *A. equolifaciens* did not influence liver steatosis induced by the high-fat, high-sugar diet. Plasma cholesterol or ferritin was not modified by *A. equolifaciens*.

Upon sacrifice, fecal content was collected and weighted. The gavage with *A. equolifaciens* resulted in a reduction of the full caecum weight, associated with an increase but statistically non-significant increase in total short-chain fatty acids (SCFA). No differences were observed in the amount of acetate and propionate. However, an important increase in the amount of butyrate was observed as well as a reduction in the proportion of the two branched-chain fatty acids, iso-butyrate and iso-valerate (Figure 9).

Finally, both the caecum wall and liver were collected to measure the level of expression of interleukin-6 (IL-6), a pro-inflammatory cytokine by RT-qPCR. In both tissues, the gavage with *A. equolifaciens* reduced the expression of IL-6, although the effect was stronger on the caecum than on the liver (Figure 10).

## 3. Discussion

### 3.1. Decreased Abundance of A. equolifaciens in the Microbiome of NAFLD/NASH Patients

Based on the decreased abundance in patients with liver diseases, we can hypothesize that *A. equolifaciens* plays a key protective role in the healthy state, as shown for *Faecalibacterium prauznitzii* in patients with inflammatory bowel diseases and for *Akkermancia muciniphila* in patients with metabolic diseases such as obesity and diabetes. Although the latter are dominant bacteria in the human microbiome, others, such as *Christensenella minuta* or *A. equolifaciens*, are sub-dominant but essential micro-organisms for the gut ecosystem [35,36,37]. Interestingly, although we focused our work on liver diseases, *A. equolifaciens* is also depleted in other pathologies including ulcerative colitis [38,39,40].

### 3.2. A. equolifaciens Presence in Human Gut

At the time of writing, *A. equolifaciens* comprises two species, the original one represented by the DSM 19450 strain and a new one named *A. rubneri*, defined by the strain ResAG-91 (DSM 111416) also called *A. hattorii* defined by the strain 8CFCBH1T (DSM 112284) [33,34]. In addition, the original species was divided into two subspecies [41]: *A. equolifaciens subsp*. *equolifaciens* and *A. equolifaciens subsp. celata*, represented by the strains DSM19450 and DSM18785, respectively [29,42]. Yet, our genomic analysis revealed that the genomes of these strains are closely related (ANI = 97.2%), and the existence of such subspecies is not supported after considering dozens of other genomes. Interestingly, although *A. equolifaciens* is named due to its capacity to transform isoflavonol into equol, metabolic potential predictions based on the presence of genes coding for key enzymes showed that the equol production pathway is absent in genomes of many strains. In fact, most strains of the Asian phylogroup are equol producers, while this metabolic capacity is rare in the phylogroup associated with North Americans and Europeans (Figure 5 and Appendix A). It is noteworthy that most bacteria from the *Coriobacteriia* class have metabolic capacities regarding polyphenols, independently of equol production; thus, this functionality may be of major interest for therapeutic purposes. However, considering the link between *A. equolifaciens* and NAFLD, we may consider that the equol-producing function is probably not essential, as we found that this bacterium was not only enriched in Asian healthy controls but in Europeans as well. Moreover, in our mouse experiments, no isoflavonol was added to the diet; thus, equol could not be the essential anti-inflammatory metabolite. In contrast, in multiple sclerosis, the protective effect provided by the association of isoflavonol and *A. equolifaciens* is due to equol production [43].

### 3.3. A. equolifaciens Presents Anti-Inflammatory Properties

Our in vitro data show that *A. equolifaciens* pellet and, to a lower extent, its supernatant, possess interesting anti-inflammatory properties through the inhibition of the NF-κB pathway, both in human intestinal epithelial cells and in hepatocytes. These properties are not present for the closely related species, *P. caecicola*, used as the negative control. Here, we showed the results obtained with the strain DSM 19450; however, we also tested about 12 other strains of *A. equolifaciens* belonging to the 2 different phylotypes and found that all strains have the same properties. Similar in vitro anti-inflammatory properties have been observed for *F. prauznitzii* on the same reporter cell line [35]. However, it was mostly caused by secreted compounds, including butyrate [44]. For *A. equolifaciens*, the effect of the supernatant is not due to butyrate (or propionate) production since they are not produced by this microbial strain. However, we cannot rule out the role of SCFA in the in vivo model since we observed an important increase in butyrate and a decrease in branched SCFA production, iso-butyrate, and isovalerate. Interestingly, the increased production of iso-butyrate and isovalerate has been found in the caecum of mice prone to NAFLD development compared to mice resistant to NAFLD [27].

Considering the in vivo mouse model, we observed an interesting effect on weight gain and glycemia at 1 month. Our PCR analysis of the fecal microbiota of our mice force fed with *A. equolifaciens* showed different abundance from animal to animal at day 55 (Appendix A) but almost complete absence at the time of sacrifice. Remarkably, our data show lower glycemia in these mice at month 1 and no significant difference at sacrifice. The absence of detectable bacteria by PCR at sacrifice may explain why no major effect was observed at that period, besides the reduction in IL-6 cytokine expression.

Considering the in vivo mouse model, we observed a clear effect on weight gain and glycemia at 1 month but not at the end of the experiment. Interestingly, PCR analysis of the fecal microbiota of mice force fed with *A. equolifaciens* performed at day 55 showed a variable abundance of the addition between different animals (Appendix A), while at the time of sacrifice, there was an almost complete absence of *A. equolifaciens*. This mirrored the return to normal weight and serum glucose at the time of sacrifice. The disappearance of this bacterial strain may explain why the effect was not sustained at this later time point.

Thus, in conclusion, *A. equolifaciens* is a particularly interesting commensal bacterium, especially in view of its anti-inflammatory effects on intestinal and hepatic cells and its action on metabolism. Even if further studies are necessary, it could represent, alone or in combination, a live biotherapeutic product candidate for the treatment of steatotic diseases, the epidemiology of which is worrying, and therapeutic options are scarce. This also warrants further investigations into other diseases including IBD.

## 4. Materials and Methods

### 4.1. Metagenome-Wide Association Study

#### 4.1.1. Study Population NASH1 Cohort

In a discovery study, 96 patients (56% men; mean BMI 29.6 kg/m^2^) with suspected NAFLD were recruited from November 2011 to January 2012 in the Department of Hepatology of the Hospital Pitié-Salpêtrière. Patients younger than 70 years old, with a liver biopsy performed within the last 3 years or having a scheduled liver biopsy during the period study, with a consent form signed, were included. Patients with incomplete or non-exploitable anamnestic and/or histological data, with bariatric surgery or recent colonoscopy, with antibiotic or probiotic intake, or with daily consumption of alcohol were excluded. Patients were stratified into three groups of increasing histological severity: normal/fatty liver alone (FL, *n* = 29), steatohepatitis with no/minimal fibrosis (*n* = 34), and steatohepatitis with bridging fibrosis/cirrhosis (*n* = 33) based on a liver biopsy performed prior (median 33 months) to the stool collection.

#### 4.1.2. Study Population NASH2 Cohort

In a validation study, 260 patients (40.4% of men; mean BMI 34.9 kg/m^2^) with suspected NAFLD were recruited from December 2012 to July 2013. Patients younger than 70 years old, with a liver biopsy performed within the last year or having a scheduled liver biopsy during the period study, with a consent form signed, were included. Patients with incomplete or non-exploitable anamnestic and/or histological data, with bariatric surgery or recent colonoscopy; other causes of liver disease such as viral, auto-immune, genetic, drug-induced steatohepatitis (methotrexate, tamoxifen, corticotherapy, amiodarone, and diltiazem); with a situation at risk of change in intestinal microbiota; chronic enteropathy with immunological or infectious components; treatment with antibiotics that had been discontinued less than 8 weeks prior to samples collection; daily probiotics intake in the last 3 weeks prior to samples collection; long-term treatment with immunosuppressant drugs; or colonoscopy within the last 3 months were excluded. Patients were stratified into four groups of increasing histological severity: normal liver (<5% of steatosis, *n* = 25), simple steatosis/fatty liver (>5% of steatosis, *n* = 52), steatohepatitis with no/minimal fibrosis (*n* = 134), and steatohepatitis with bridging fibrosis/cirrhosis (*n* = 49) based on a liver biopsy performed prior to the stool collection.

#### 4.1.3. Reference Healthy Volunteer Populations Spanish MetaHIT

As a reference healthy population, 91 metagenomics samples were used from the MetaHIT cohort [45] (NCBI = PRJEB1220). The healthy volunteers were adults and from Spain.

#### 4.1.4. Analysis of MGS in NASH1, NASH2, and Healthy Cohort (MetaHit)

Patient samples were procured, collected, stored, and disseminated in accordance with the highest ethical standards and in the strictest compliance with all applicable rules and regulations. This includes ensuring that consents were fully informative for donors and that the donor’s wishes in relation to the use of his/her samples were strictly complied with. Fecal microbiota analysis was performed according to previously used methodology [46,47]. DNA was extracted from the stool samples, sequenced using SOLiD™ technology, and mapped to the MetaHIT reference catalog. Quantitative metagenomics analysis consisted of microbial gene count and identification of 900 MetaGenomic Species (MGS = clusters of >500 genes that co-vary at the same level of abundance from one sample to another) [45].

#### 4.1.5. Data Availability

Shotgun metagenomic data of public cohorts were downloaded from the European Nucleotide Archive. Full list of samples used in this study is available in Appendix A.

#### 4.1.6. Sequencing Data Preprocessing

Sequencing data quality control was performed with fastp [48] (parameters: --cut_front --cut_tail --n_base_limit 0 --length_required 60) to remove low-quality reads and sequencing adapters. Then, reads mapped to the human genome (T2T CHM13v2.0 GCA_009914755.4) with bowtie2 [49] were removed. Finally, read subsampling was performed with fastq-sample (github.com/fplaza/fastq-sample) to take into account variable sequencing depth across samples. The number of reads subsampled in each sample is indicated in Appendix A.

#### 4.1.7. Gene Coverage Table Generation

The gene abundance table was generated with the METEOR software suite (forgemia.inra.fr/metagenopolis/meteor (accessed on 27 May 2023)). First, selected high-quality reads from each sample were mapped with bowtie2 [49] against the updated Integrated Gene Catalogue of the human gut microbiome [50] (IGC2, 10.4 million genes). Alignments with nucleotide identity ≥ 95% were kept to account for intra-species nucleotidic variability and the non-redundant nature of the catalog. Then, raw gene counts were computed with a two-step procedure previously described that handles multi-mapped reads [51]. Finally, gene coverage was computed by dividing raw gene counts by gene length multiplied by average read length in sample.

#### 4.1.8. Species-Level Taxonomic Profiling

Using MSPminer [52], the IGC2 catalog was previously organized into 1990 MetaGenomic Species (MGS), which were clusters of co-abundant genes corresponding to the same microbial species [53]. The MGS named msp_0396 was identified as representative of *A. equolifaciens*. The abundance of this MGS in a sample was defined as the mean coverage of its 100 marker genes (i.e., species-specific core genes that correlate the most altogether). If less than 10% of the marker genes was seen in a sample, the abundance of the MGS was considered as null. The estimated sequencing coverage of *A. equolifaciens* (msp_0396) across samples is reported in Appendix A.

#### 4.1.9. Statistical Analysis

Statistical analysis was performed using R software suite. MGS differentially abundant between controls and patients were identified using Mann–Whitney U tests.

### 4.2. Population Genomics

#### 4.2.1. Data Availability

A total of 154 high-quality Metagenome-Assembled Genomes (MAGs) (completeness ≥ 80% and contamination ≤ 5% assessed with CheckM2 [54]) of *A. equolifaciens* were downloaded from various sources including the following: (1) the Unified Human Gastrointestinal Genome collection (UHGG) [55], (2) the Human Reference Gut Microbiome (HRGM) [56], and (3) the Murine Intestinal Microbiota Integrated Catalog v2 (MIMIC2) [57]. In addition, 11 genome sequences derived from cultured isolates were downloaded from the NCBI GenBank (September 2022). Full list of genomes used in this study is available in Appendix A.

#### 4.2.2. Bioinformatics and Biostatistics

Average Nucleotide Identity (ANI) between all pairs of genomes was computed with fastANI [32]. A 165 × 165 distance matrix was built where distance was defined as “1-ANI”. Then, this matrix was submitted to a Non-Metric Multidimensional Scaling (NMDS) process implemented in vegan R package [58]. The NMDS output was used to map the genomes in a two-dimensional plot. Finally, genomes were clustered into phylogroups using average linkage hierarchical clustering (UPGMA) using a 95% ANI cut-off.

Sequences of proteins involved in the equol production pathway were downloaded from the NCBI: daidzein reductase (dzr) = WP_022741749.1, dihydrodaidzein reductase (ddr) = WP_022741751.1, tetrahydrodaidzein reductase (tdr) WP_022741752.1, and WP_022741755.1 (racemase). These four proteins were searched in the genomes with tblastn [59] (amino acid identity ≥ 90%, query coverage ≥ 90%). A strain was considered as an equol producer if at least 75% of the proteins was detected in its genome.

### 4.3. In Vitro Experiments

#### 4.3.1. Cell Culture and Reagents

HepG2 human hepatocarcinoma cells and HT-29 human intestinal epithelial cells were grown in Dulbecco’s Modified Eagle’s Medium (DMEM, Gibco, Les Ullis, France) and Roswell Park Memorial Institute (RPMI, Gibco, Les Ullis, France) 1640, respectively, and supplemented with 2 mM L-glutamine (Gibco, Les Ullis, France), 50 IU/mL penicillin (Gibco, Les Ullis, France), 50 µg/mL streptomycin (Gibco, Les Ullis, France), and 10% heat-inactivated fetal calf serum (Eurobio, Les Ulis, France). DMEM was also supplemented with 1% nonessential amino acid (NEAA) 100X (Gibco, Les Ullis, France), 1% hepes 1M (Gibco, Les Ullis, France), and 1% sodium pyruvate 100 mM (Gibco, Les Ullis, France). Cells were cultures in a humidified 5% CO2 atmosphere at 37 °C. Absence of mycoplasma contamination was controlled using a MycoAlert kit (Lonza, Levallois-Perret, France).

Construction and validation of the NF-κB reporter clones HepG2/κb-SEAP was conducted following the same protocol as the one used to generate HT-29 NF-κB-SEAP reporter cell lines and described previously [60].

#### 4.3.2. Commensal Strains and Preparation of Conditioned Media

*A. equolifaciens* (DSM 19450) and *P. caecicola* (DSM 22242) bacteria were grown in anaerobic M104 medium supplemented with 0.5% arginine at 37 °C following the Hungate culture method [61]. After overnight incubation, bacterial cultures were centrifuged at 11,000 rpm for 10 min. The pellets and supernatants were collected separately, and the supernatant was filtered on 0.2 µm PES filters (Corning, NY, USA) before collection. The samples were stored at −80 °C until their use. Non-inoculated bacteria culture medium served as the control. For the experimentation, cell pellets were resuspended in PBS (Gibco, Les Ullis, France).

#### 4.3.3. Analyses of NF-κB Activation

For each experiment, reporter cells were seeded at 30,000 cells per well, into 96-well plates and incubated for 24 h. Then, cells were stimulated for 24 h with 10 µL of tested bacteria supernatant or resuspended bacterial pellet, for a final volume of 100 µL per well (i.e., 10% vol/vol), in the presence or absence of TNF-α (10 ng/mL final). Secreted Embryonic Alkaline Phosphatase (SEAP) in the supernatant was revealed using the Quanti-Blue^TM^ reagent (Invivogen, Toulouse, France) according to the manufacturer’s protocol and quantified at 655 nm OD. All measurements were performed using a microplate reader (Infinite 200, Tecan, Lyon, France). Cell viability was controlled using an MTS assay (CellTiter 96 Aqueous One, Promega, Charbonnières-les-Bains, France) following the manufacturer’s instructions.

#### 4.3.4. Statistical Analysis

Results were expressed as the mean ± the standard error of the mean (SEM) of three independent experiments with triplicate determinations. The statistical analysis was performed using GraphPad Prism version 8 (La Jolla, CA, USA). One-way ANOVA was used followed by Dunnet’s multiple comparisons test. *p*-values < 0.5 were considered as significant.

### 4.4. Animal Experiments

#### 4.4.1. Clinical Cohort for Stool Selection

A cohort of 20 NAFLD patients with moderate obesity, aged 62 on average, was recruited; a liver biopsy determined a diagnosis of NAFLD or NASH. All subjects gave their informed consent for inclusion before they participated in the study. The study was conducted in accordance with the Declaration of Helsinki, and the protocol was approved by the French CPP Ethics Committee. A donor with NASH was selected among the 20 donors.

#### 4.4.2. Preparation and Preservation of Fecal Transplants

Stools were collected and stored at −80 °C in a preservation solution in MD diluent kindly provided by MaaT Pharma (Lyon, France). Fecal transplant was prepared as previously described [62].

#### 4.4.3. Animal Experimentation

Procedures were performed according to the European Guidelines for the Care and Use of Laboratory Animals and approved by the French Veterinary Authorities (authorization number 78–60). The experimental protocol was agreed by the French Ministère de l’Education Nationale, de l’Enseignement Supérieur et de la Recherche (APAFIS#18425-2018110521086756 v2). Twenty-four specific pathogen-free (SPF) C57Bl/6J male mice, 7 weeks old, were purchased from Charles River Laboratories (Misery, France) from different litters. On arrival, animals were housed under controlled conditions of temperature, hygrometry, and 12 h light/dark cycle in an SPF animal facility (Infectiologie Expérimentale des Rongeurs et des Poissons, IERP) at INRAE, Jouy-en-Josas. They were individually weighted, microchipped, and randomly housed in new cages, 4 mice per cage; they received a conventional gamma-irradiated 45 kGy mice control diet (CD) SAFEA03 R03-40 ad libitum (3.24 kcal/g: 14% energy from fat; 25% energy from proteins; 61% energy from carbohydrates; SAFE, Augy, France, CD).

On day 6 after arrival, feces of the mice were individually harvested (basal microbiota). Then, they received in the autoclaved drinking water for 2 weeks a mixture of broad-spectrum antibiotics (vancomycin 45 µL/mL, streptomycin 1 mg/mL, colistin 1 mg/mL, and ampicillin 1 mg/mL) ad libitum to clean their endogenous microbiota, as previously described [63]. On day 16, an all-bacteria-qPCR was performed to confirm complete absence of detectable bacteria in the feces (threshold of 99.99% depletion was considered to be equivalent to germ-free mice). The stools of the selected human NASH patient were inoculated (200 µL per mouse) by two gavages at 48 h intervals (day 21 and 22). Fecal transplants contained more than 70% of viable bacteria as shown by flow cytometry. Mice were fed ad libitum for 10 weeks a high-fat, high-fructose diet (2HFD) to induce NAFLD following a protocol previously described [28]. The first group of 12 mice was gavaged five times a week from day 27 until euthanasia, i.e., 50 days, with sterile PBS. A second group of 12 mice was gavaged with resuspension of the dry pellet of *A. equolifaciens* diluted in PBS. Body weight, food, and liquid consumption were monitored weekly. Mice stools were collected individually at five time points: on day 6 (basal), 16 (after 2-week antibiotics treatment), 27 (1 week after fecal microbiota transplant), 55 (1 month after fecal microbiota transplant), 84 (1 week before the end of the experiment). On day 90, mice were euthanatized, and liver, epididymal and mesenteric adipose tissue, caecum, and blood were then harvested.

#### 4.4.4. Preparation of *A. equolifaciens* for Gavage

After rehydration of the lyophilized vial with 500 μL of JCM 663 rehydration medium, 100 μL were spread in a Petri dish on Wilkins Chalgren anaerobic agarose medium containing 0.5% arginine and incubated for 3 days at 37 °C in a Freter chamber. Two pre-cultures were performed in order to obtain 64 Petri dishes. For each dish, bacteria were resuspended in sterile PBS and aliquoted in 2 mL tubes at an optical density of one.

#### 4.4.5. 16S rRNA Sequencing Analysis

DNA was extracted from feces or caecum content using Gnome DNA Isolation Kit (MP Biomedicals, Santa Ana, CA, USA). Quantifications of the total bacteria DNA were performed by real-time qPCR following the procedure previously described [64]. The V3-V4 region of the 16S rRNA genes was amplified using MolTaq (Molzym, Bremen, Germany) with the primers described previously [65]. Purified amplicons were sequenced using the MiSeq sequencing technology (Illumina, San Diego, CA, USA) at the GeT-PLaGe platform (Genotoul, Toulouse, France). Paired-end reads obtained from MiSeq sequencing were analyzed using the Galaxy-supported pipeline named find, rapidly, operational taxonomic units (OTUs) with Galaxy Solution (FROGS) [66]. For the preprocessing, reads with length between 380 bp and 500 bp were retained. The clustering and chimera removal tools, followed the guidelines of FROGS. Assignment was performed using the Silva 132 database updated in December 2017 with top quality pintail 100 (https://www.arb-silva.de/, accessed on December 2017). OTU with abundances lower than 0.005% were removed from the analysis [67]. 16S sequencing data were analyzed using the Phyloseq, DESeq2 and ggplot2 R packages in addition to custom scripts as described previously [68].

#### 4.4.6. Short-Chain Fatty Acids Quantification of Cecal Contents

Measurement of the short-chain fatty acids (SCFAs) and branched-chain fatty acids from mice cecal contents was performed using gas chromatography, as previously described [69]. The analyses were performed on an Agilent (Les Ulis, France) gas chromatograph equipped with a split–splitless injector 7850 and ionization flame detector. Carrier gas (H2) flow rate was 10 mL/min, and inlet, column, and detector temperatures were 200, 100, and 240 °C, respectively. Data were collected and peaks were integrated using OpenLabChem station software (Les Ulis, France).

#### 4.4.7. Plasma Assays

On day 90, blood was collected from mice before euthanasia by submandibular puncture into tubes containing 5 µL EDTA 0.5 mol/L. After centrifugation (6000× *g*, 20 min, 4 °C) plasma was aliquoted and frozen at −80 °C until further analysis. Measurements of plasma aspartate-transferase (AST), alanine-transaminase (ALT), triglycerides (TG), high-density lipoprotein cholesterol (HDL), total cholesterol, and ferritin were performed using the biochemistry platform (CRI, UMR 1149, Paris) with an Olympus AU400 Chemistry Analyzer. Llow-density lipoprotein (LDL) cholesterol was calculated according to the Friedewald formula: LDL-cholesterol = total cholesterol − HDL-cholesterol − TG/2.2 (mmol/L), with all TG < 4.6 mmol/L.

Measurements of non-fasting plasma insulin and leptin were performed using mouse-specific insulin and leptin ELISA Kit (Merck Millipore, Molsheim, France).

#### 4.4.8. Real-Time Quantitative Polymerase Chain Reaction (qPCR)

A 1/4 portion of the left liver lobe was stored in RNAlater^TM^ stabilization solution (Invitrogen, Carlsbad, CA, USA) at −80 °C until further analysis. Total RNA was extracted from the liver with RNAeasy Plus Mini Kit (Qiagen, Hilden, Germany). RNA integrity and concentration were checked with RNA 6000 Nano chips on an Agilent 2100 bioanalyser (Agilent Technologies, Amsterdam, the Netherlands). Total RNA (10 µg per reaction) was reverse transcribed into complementary DNA using high-capacity cDNA reverse transcription kit (Applied Biosystems, Thermofisher Scientific, Foster City, CA, USA) according to the manufacturer’s instructions. Real time qPCR was performed on an Applied Biosystems Step One Plus machine. The relative gene expressions were normalized to two housekeeping genes: *Gapdh* and *Actb* (glyceraldehyde-3-phosphate dehydrogenase, actin beta) chosen on the basis of results obtained from TaqMan mouse endogenous control arrays (Applied Biosystems).

#### 4.4.9. Oral Glucose Tolerance Test (OGTT)

Fasting glycemia and insulinemia measurements as well as OGTT were performed on day 70, i.e., 7 weeks after the 2HFD regimen and 21 days before euthanasia. After 6 h of fasting, a glucose solution (2 g glucose/kg body weight) was administered by oral gavage. Blood glucose levels at time 0 (fasting glycemia, determined before glucose gavage) and 15, 30, 60, 90, and 120 min after glucose gavage were analyzed using an Accu-Check glucometer (Roche, Meylan, France). The glucose levels were plotted against time, and the AUC (area under curve) was calculated. The plasma insulin concentrations at times 0 (fasting insulinemia) and after 30 min, were analyzed in venous blood (collected in EDTA-coated tubes), harvested from marginal tail vein, using a mouse-specific Insulin ELISA Kit (Merck Millipore, St Quentin en Yvelines, France). Insulin resistance was estimated by homeostasis model assessment (HOMA-IR index) and calculated according to the following formulas: HOMA-IR (fasting) = fasting glucose (mmol/L) × fasting insulin (mU/L)/22.5. HOMA-IR (non-fasting) = non-fasting glucose (mmol/L) × non-fasting insulin (mU/L)/22.5.

#### 4.4.10. Statistical Analysis

Datasets normality was tested using the D’Agostino and Pearson normality test. Normally distributed data with equal group variances were expressed as mean ± standard errors of the mean (SEM). Non-normally distributed data, or belonging to unequal group variances, were expressed as medians (interquartile ranges). The level of significance was set at *p* < 0.05 (* *p* < 0.05, ** *p* < 0.01, *** *p* < 0.001). Calculations were performed with R 3.5 software and GraphPad Prism software (version 7.00, La Jolla, CA, USA).

## 5. Patents

Based on this work, a patent was deposited under the application number PCT/EP2021/073417.

## Figures and Tables

**Figure 1 ijms-24-12232-f001:**
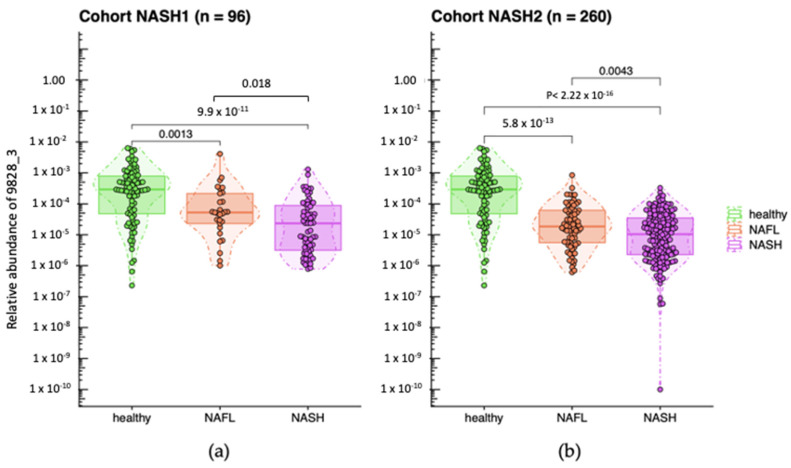
Relative abundance of *A. equolifaciens* (MGS 9828_3) in groups of patients determined according to their clinical status based on biopsy readings in NASH1 (**a**) and NASH2 (**b**) cohorts. Healthy: healthy controls; NAFL: patients with steatohepatitis with no/minimal fibrosis; NASH: patients with steatohepatitis with and without fibrosis or cirrhosis. Relative abundances are log10-transformed, and “M10” is an artificial value introduced when the species was not detected. *A. equolifaciens* was significantly (Wilcoxon rank sum tests) more abundant in healthy volunteers compared to NASH/NAFL populations (NASH1: *p* < 0.0001; NASH2: *p* < 0.0001). Moreover, *A. equolifaciens* was more abundant in patients with NAFL compared to the NASH population (NASH1: *p* = 0.018; NASH2: *p* = 0.0043).

**Figure 2 ijms-24-12232-f002:**
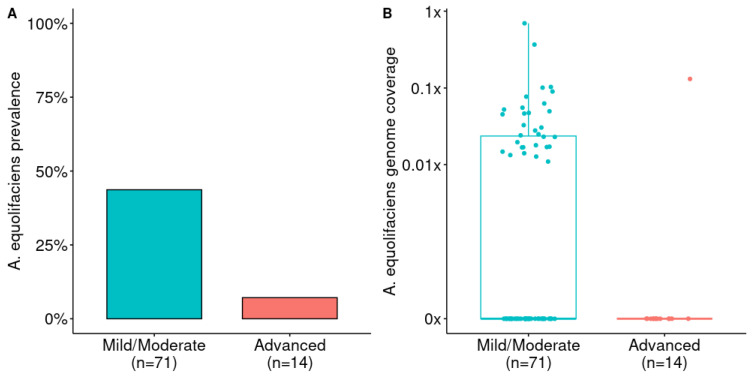
(**A**) Prevalence of *A. equolifaciens* in patients with NAFLD depending on the level of fibrosis. Analysis from publicly available metagenomic data from Loomba et al. [10]. Fisher’s exact test: *p* = 0.014. (**B**) Relative abundance of *A. equolifaciens* in patients with NAFLD depending on the level of fibrosis. Mann–Whitney U test: AUC = 0.67 and *p* = 0.023.

**Figure 3 ijms-24-12232-f003:**
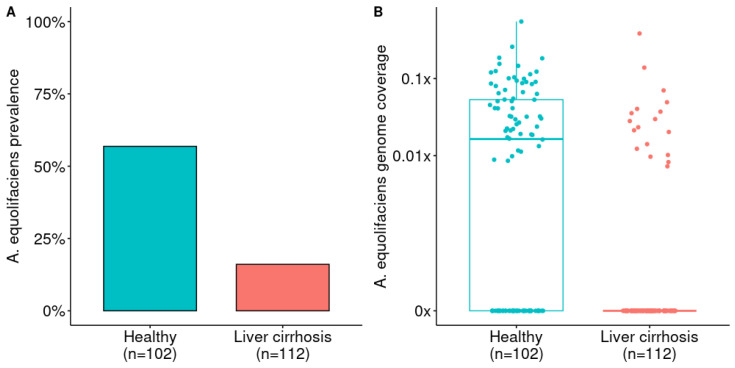
(**A**) Prevalence of *A. equolifaciens* in patients with liver cirrhosis compared to healthy controls. Analysis from publicly available metagenomic data from Qin et al. [30]. Fisher’s exact test: *p* = 3.91 × 10^−10^. (**B**) Relative abundance of *A. equolifaciens* in patients with liver cirrhosis compared to healthy controls. Mann–Whitney U test: AUC = 0.72 and *p* = 9.73 × 10^−11^).

**Figure 4 ijms-24-12232-f004:**
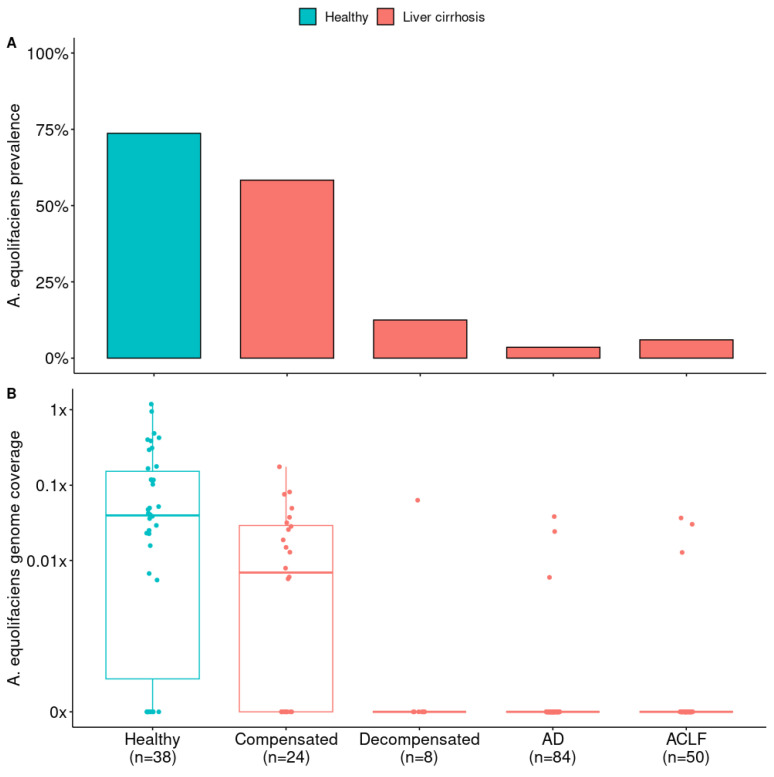
(**A**) Prevalence of *A. equolifaciens* in patients with liver cirrhosis according to disease severity (Compensated, Decompensated, Acute Decompensation, and Acute-on-Chronic Liver Failure) and compared to healthy controls. Analysis from publicly available metagenomic data from Sole et al., 2021 [31]. Healthy controls versus all liver cirrhosis patients, Fisher’s exact test: *p* = 2.39 × 10^−13^. Liver cirrhosis patients only according to disease severity, chi-squared test: *p* = 1.37 × 10^−11^. (**B**) Abundance of *A. equolifaciens* in patients with liver cirrhosis according to disease severity and compared to healthy controls. Healthy controls versus all liver cirrhosis patients, Mann–Whitney U test: AUC = 0.83; *p* < 2.2 × 10^−16^. Liver cirrhosis patients only according to disease severity, Kruskal–Wallis test: *p* = 1.66 × 10^−11^.

**Figure 5 ijms-24-12232-f005:**
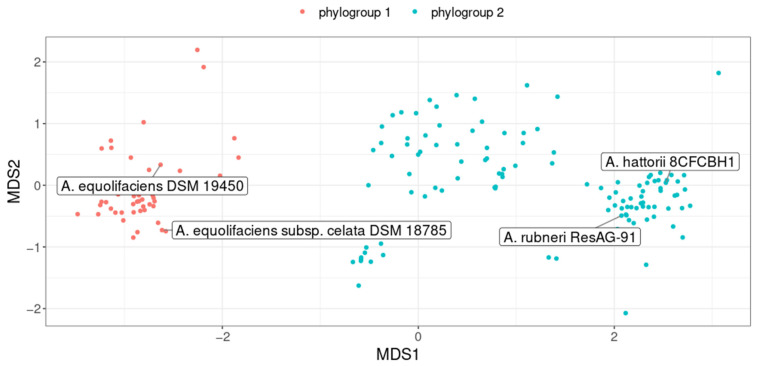
Non-Metric Multidimensional Scaling (nMDS) plot representing genomes similarity assessed from pairwise ANI. Each point represents a genome and the color indicates the phylogroup (genomospecies) to which it belongs. Genomes of the four type strains are labeled.

**Figure 6 ijms-24-12232-f006:**
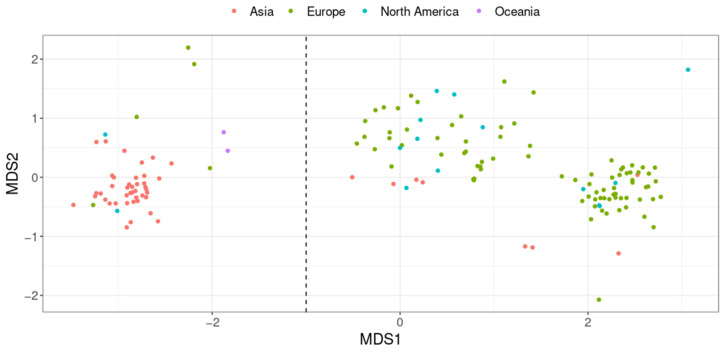
Non-Metric Multidimensional Scaling (nMDS) plot representing genome similarity assessed from pairwise ANI. Each point represents a genome, and the color indicates the geographical origin of the host. The vertical dashed line separates the two phylogroups (genomospecies). Non-human-related genomes are not shown.

**Figure 7 ijms-24-12232-f007:**
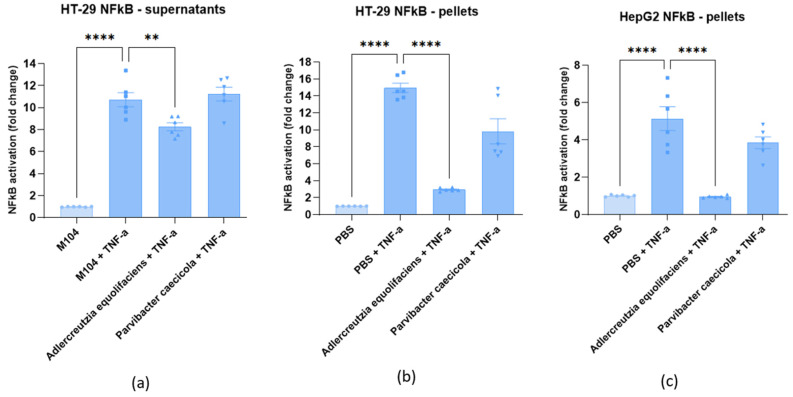
Effect of supernatant and pellet of *A. equolifaciens* and *P. caecicola* on NF-κB activity of HT-29 (**a**,**b**) and HepG2 (**c**) reporter cell lines. NF-κB activity was measured in the absence (first bar) or presence of TNF-α. Values represent fold increase in NF-κB activity over unstimulated cells (M104 or PBS). Results are expressed as the mean ± SEM of at least three experiments with triplicate determinations. *p*-values < 0.05 are considered significant ** *p* < 0.0021, **** *p* < 0.0001).

**Figure 8 ijms-24-12232-f008:**
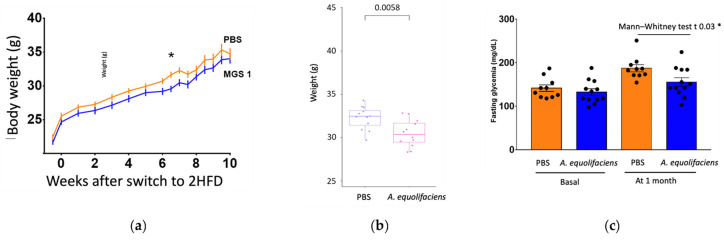
(**a**) Average weight (+/− SEM) along time of mice receiving *A. equolifaciens* (MGS1) or PBS as control. (**b**) Weight of mice at the end of the experiment. (**c**) Fasting glycemia of mice at the beginning of the experiment (basal) and 1 month after starting the gavage with PBS or *A. equolifaciens*. *n =* 12 for each group, * *p* < 0.05.

**Figure 9 ijms-24-12232-f009:**
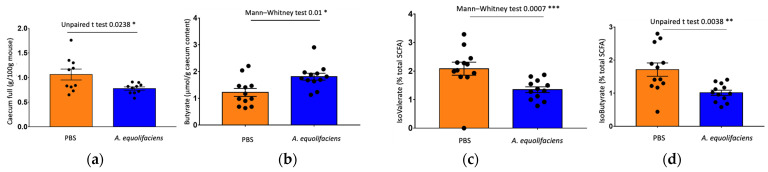
*A equolifaciens* gavage modulates caecum weight and caecal SCFA concentration. (**a**) Weight of caecum/mouse weight at the end of the experience. (**b**) Butyrate concentration in μmol/g of caecum content. (**c**) D-Iso-valerate concentration/g of caecum content expressed as % of total SCFA. (**d**) Iso-butyrate concentration/g of caecum content expressed as % of total SCFA. Orange—mice receiving PBS; blue—mice receiving *A. equolifaciens*. *n =* 12 for each group, * *p* < 0.05, ** *p* < 0.01, *** *p* < 0.001.

**Figure 10 ijms-24-12232-f010:**
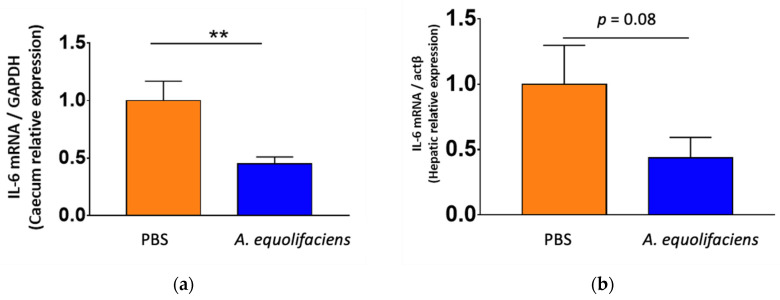
*A. equolifaciens* gavage reduces IL-6 mRNA expression in the caecum (**a**) and the liver (**b**) of mice. GAPDH and actin-β were used as housekeeping genes in the caecum and the liver, respectively. *n* = 12 for each group, ** *p* < 0.01.

## Data Availability

The data presented in this study are available on request from the corresponding author unless specified in the Material and Methods section.

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
