# Peer review of "Adlercreutzia equolifaciens* Is an Anti-Inflammatory Commensal Bacterium with Decreased Abundance in Gut Microbiota of Patients with Metabolic Liver Disease"

_ijms, 2023, doi:10.3390/ijms241512232_

Round 1
Reviewer 1 Report
The paper " Adlercreutzia equolifaciens is an anti-inflammatory commensal bacterium with decreased abundance in gut microbiota of patients with metabolic liver disease" presents the problem of NAFLD and the possibility to find the reason in partial why this methabolic problem is presented.
The paper is well designed but some parts should be improved.
The introduction is very long. It should be shorten.
What Authors wanted to say by "shotgun" metagenomic?
The paper is full of experiments, just it is unusual to use the data from 2011. Those data are from whom?
English must be reviewed. Don't use Present continious tense in paper. It is not correct.
Section 4.4 with title Animal studies, and than talking about 4.4.1 cochort studies is not correct.
Be careful about units. The must be uniformed. Sometimes thery are next to number, somethimes they are with space. Celsious must be written with capital letter (line 523)
There is very little explanation about fatty chain acids composition. There is no conclusion about it. It is not clear what part of the paper has a patent. be specific.The English must be imporved by an English speaker who understand the research
Author Response
We thank the reviewer for the helpful comments.
Considering comment 1 : we believe that every paragraph of the introduction is needed for the comprehension of the paper and decided not to shorten it in a concern to remain comprehensive.
Considering comment 2 : Shotgun metagenomic is the classical method to study gut microbiota and this term is used for such studies. Full details of the method are described in the m&m section.
Considering comment 3 : Data from 2011 are needed because it is the beginning of the story, those data are from some of the authors.
Considering comment 4 : The english has been reviewed and Present is now only used when necessary.
Considering comment 5 : We modified the title of section 4.4.1. to make it clear. Indeed for the animal model, a human cohort was needed to select the proper donor for the animal study.
Considering comment 6 : We tried to uniform the units and corrected the required word.
Considering comment 7 : we added a sentence on SCFA.
Considering the last comment : the patent which is published relies on the in vitro and animal work and is now published. The sentence has been modified.
Again we thank the reviewer for all helpful comments.
Reviewer 2 Report
The topic of the manuscript is consistent with the scope of the journal. The authors explore a biotherapeutic strategy using Adlercreutzia equolifaciens for liver diseases. The topic is current and well described. I have only a comments:
Comment 1: In all statistical analysis topics, include the level of significance and symbols used, distribution analysis and program used,
Comment 2: In all legendas add the n used.
Comment 3: Standardize the images.
Author Response
Considering comment 1 : Statistics and symbols are presented in the M&M section, when not in the figures themself.
Considering comment 2 : for each figure the n used is now presented either in the figure itself or in the legend.
Considering comment 3 : standardisation is done for each of the 3 parts.
Reviewer 3 Report
This study is very important because it examines the role of gut microbiota ( Adlercreutzia equolifaciens bacterium) in the NAFLD pathogenesis with shotgun metagenomics. This is complex investigation This is a complex investigation that includes clinical studies, as well as experimental ones (in vitro and in vivo). The study was conducted in accordance with the Declaration of Helsinki, and the protocol was approved by the French CPP Ethics Committee. For animal use procedures were performed according to the European Guidelines for the Care and Use of Laboratory Animals and approved by the French Veterinary Authorities. The experimental protocol was agreed upon by the French Ministère de l’Education Nationale, de l’En seignement Supérieur et de la Recherche (APAFIS#18425-2018110521086756 v2).
DNA of feces microbiota of was extracted from the stool samples, sequenced using SOLiD™ technology, and mapped to the MetaHIT reference catalog. In two studies, (Study population NASH1 cohort, discovery study, and Study population NASH2 cohort, validation study), the authors analyzed the gut microbiota with shotgun metagenomics. Quantitative metagenomics analysis consisted of microbial gene count and identification of 900 MetaGenomic Species. The authors use in vivo (mice with NAFLD) and in vitro (Cell culture, HepG2 human hepatocarcinoma cells).
A. equolifaciens is a commensal bacterium, with anti-inflammatory effects and its action on the metabolism. The authors show that there is a link between NAFLD and the severity of liver disease and the presence of A. equolifaciens and its anti-inflammatory actions. They believe that gut dysbiosis in NAFLD with this bacterium may be a promising live biotherapeutic strategy for metabolic liver diseases.
Based on the significance of the research, the complex design of the study, the complex methodology, as well as and the results obtained, which are clearly presented in the figures, I think that this article is suitable for publication in IJMS.
Minor comment:
In Paragraph 4.1.2 Study Study population NASH2 cohort - delete study
Author Response
Thanks for the kind comments. Your minor correction has been corrected.